# Whole Exome Sequencing Reveals Novel Candidate Genes in Familial Forms of Glaucomatous Neurodegeneration

**DOI:** 10.3390/genes14020495

**Published:** 2023-02-15

**Authors:** Kiran Narta, Manoj Ramesh Teltumbade, Mansi Vishal, Samreen Sadaf, Mohd. Faruq, Hodan Jama, Naushin Waseem, Aparna Rao, Abhijit Sen, Kunal Ray, Arijit Mukhopadhyay

**Affiliations:** 1Genomics & Molecular Medicine, CSIR-Institute of Genomics & Integrative Biology, Mathura Road (Near Sukhdev Vihar), New Delhi 110025, India; 2Academy of Scientific and Innovative Research (AcSIR), Ghaziabad 201002, India; 3CSIR-Indian Institute of Chemical Biology, Raja S. C. Mullick Road, Kolkata 700032, India; 4Institute of Ophthalmology, University College London, London EC1V 9EL, UK; 5L. V. Prasad Eye Institute, Bhubaneswar 751024, India; 6Drishti Pradip, Kolkata 700068, India; 7Translational Medicine Unit, Biomedical Research & Innovation Centre, University of Salford, Salford M5 4WT, UK

**Keywords:** blindness, exome sequencing, genetics, genomics, glaucoma, PACG, POAG, SRFBP1

## Abstract

Glaucoma is the largest cause of irreversible blindness with a multifactorial genetic etiology. This study explores novel genes and gene networks in familial forms of primary open angle glaucoma (POAG) and primary angle closure glaucoma (PACG) to identify rare mutations with high penetrance. Thirty-one samples from nine *MYOC*-negative families (five POAG and four PACG) underwent whole-exome sequencing and analysis. A set of prioritized genes and variations were screened in an independent validation cohort of 1536 samples and the whole-exome data from 20 sporadic patients. The expression profiles of the candidate genes were analyzed in 17 publicly available expression datasets from ocular tissues and single cells. Rare, deleterious SNVs in *AQP5, SRFBP1, CDH6* and *FOXM1* from POAG families and in *ACACB, RGL3* and *LAMA2* from PACG families were found exclusively in glaucoma cases. *AQP5, SRFBP1* and *CDH6* also revealed significant altered expression in glaucoma in expression datasets. Single-cell expression analysis revealed enrichment of identified candidate genes in retinal ganglion cells and corneal epithelial cells in POAG; whereas for PACG families, retinal ganglion cells and Schwalbe’s Line showed enriched expression. Through an unbiased exome-wide search followed by validation, we identified novel candidate genes for familial cases of POAG and PACG. The *SRFBP1* gene found in a POAG family is located within the GLC1M locus on Chr5q. Pathway analysis of candidate genes revealed enrichment of extracellular matrix organization in both POAG and PACG.

## 1. Introduction

Glaucoma is the leading cause of irreversible blindness in the world, projected to affect more than 100 million people globally by 2040 [1]. It is a complex multifactorial disorder characterized by the progressive loss of the retinal ganglion cells in the optic nerve. Glaucoma can be classified, based on the width of the irido-corneal angle in the anterior chamber, as primary angle closure glaucoma (PACG) [2] or primary open angle glaucoma (POAG) [3,4]. PACG is caused due to obstruction in the outflow of aqueous fluid due to closure of the irido-corneal angle, leading to increase in intraocular pressure. Based on age of onset, glaucoma can be classified as primary congenital glaucoma (PCG), occurring before 4 years of age; juvenile open angle glaucoma (JOAG), also called early-onset glaucoma, which occurs between ages 4 and 40 years; and the most common POAG, or late-onset glaucoma, that affects people over the age of 40 years [5]. 

The major endophenotypes of glaucoma have high heritability (POAG range 0.17 to 0.81 and 0.65 for PACG [6]), suggesting genetics plays an important role in disease biology [7]. Glaucoma is genetically heterogeneous and both rare (Mendelian/familial form) and common variations (multifactorial/sporadic form) are involved in the disease. Genetic studies in the past have identified several genes and risk loci [8]. Two POAG cohort-based studies revealed that 77.06% [9] and 92.2% [10] of the patients remained unaccounted by mutations in the known Mendelian genes. As glaucoma is known to cluster in families, genetic studies of families focusing on identifying novel genes and variations with higher penetrance are much needed, and the central rationale of the work presented here. More genetic studies are required to further understand the disease biology and identify potential treatment targets.

In this study, we have performed exome sequencing of 31 individuals from nine glaucoma families from India and have screened and prioritized genetic variations in an independent cohort of 1536 samples. We have also analyzed expression of the candidate genes in glaucoma-related studies and identified the protein networks involved in the disease pathogenesis.

## 2. Materials and Methods

### 2.1. Clinical Diagnosis and Sample Collection

The sample collection for the study was done in accordance with the tenets of the declaration of Helsinki and the institutional review boards that approved the study. The peripheral blood samples were taken only after informed consent, and the consent forms were duly filled and signed by the individuals taking part in the study.

The individuals were recruited from two different clinics in eastern India, namely (i) Drishti Pradip in Kolkata, West Bengal, and (ii) L.V. Prasad Eye Institute in Bhubaneswar, Odisha. IOP measurement was performed by Goldman applanation tonometry. The central corneal thickness measurement was performed by ultrasound pachymetry, and biometry included axial length, anterior chamber depth and lens thickness. Gonioscopy was performed by the four-mirror technique. Fundus bio-microscopy was carried out and disc imaging was done by Cirrus Spectral domain OCT and Humphrey visual fields 24-2 (or 10-2 or macular program, as indicated).

Primary open angle glaucoma patients were recruited if they had open angles on gonioscopy and were positive for 2 out of the 3 criteria, namely (i) intra-ocular pressure (IOP) > 21 mm of Hg, (ii) glaucomatous field damage and (iii) significant cupping of the optic disc.

Primary angle closure glaucoma was classified as per ISGEO guidelines, which included individuals with narrow angles (also termed PAC suspect) if at least 1 eye had narrow angles, PAC eyes having peripheral anterior synechiae in addition, or raised IOP (defined as an IOP of more than 21 mm Hg), or both, but without glaucomatous optic neuropathy [11]. Primary angle closure glaucoma was defined as eyes with PAC and glaucomatous optic neuropathy, defined as a vertical cup-to-disc ratio of 0.8 or more or a cup-to-disc asymmetry of more than 0.2 focal notching, or a combination thereof, with compatible visual field loss on static automated perimetry (Swedish interactive threshold algorithm Standard algorithm with a 24-2 test pattern; Humphrey Visual Field Analyzer II). This was defined as glaucoma hemifield test results outside normal limits with an abnormal pattern standard deviation, with *p* < 0.05 occurring in the normal population and fulfilling the test reliability criteria (fixation losses of more than 20%, false positives of more than 33%, false negatives of more than 33%, or a combination thereof).

Patients with a history of previous trauma or secondary glaucoma due to steroids or neovascular glaucoma and ocular hypertension without cupping were excluded from the study. Family history was evaluated with detailed pedigree, with affected and unaffected family members undergoing all the aforementioned procedures. For families with a positive history, a minimum of two affected members were chosen for the study. Healthy relatives from the families were also recruited, where possible. Clinical variables were compared among affected and unaffected family members of familial forms of glaucoma, including age and IOP at presentation, biometric indices (axial length, AC depth and lens thickness), and corneal thickness along with severity of disc/field damage at presentation. A set of 20 sporadic glaucoma samples were also collected.

The samples for validation belonged to an East Indian Cohort and are described in a previous study [12]. A set of 960 cases and 576 controls were ascertained at Dristi Pradip, Kolkata, for validation study. The patients were examined thoroughly for glaucomatous phenotype and those with intraocular pressure (IOP) of more than 21 mm Hg, glaucomatous field damage and significant optic disk cupping were identified. The patients satisfying any two of the three criteria were selected as cases for the study. The individuals which tested negative for glaucoma and did not have a history of eye disease were selected as controls.

### 2.2. Screening of MYOC Mutations as an Exclusion Strategy

The genomic DNA was extracted using the salting-out method [13]; briefly, the cells were separated from blood followed by cell lysis. The DNA was precipitated using chilled isopropanol and washed using 70% ethanol. The DNA was stored in TE Buffer at −20 °C. Before proceeding for exome sequencing, the probands in all families were screened for variations in *MYOC* using Sanger sequencing. *MYOC* is the most frequently mutated gene identified for POAG, especially in familial cases, accounting for more than 5% of the cases [14,15,16]. Mutations in this gene have also been reported in PACG [17]. As exon1 and exon3 harbors most of the known mutations, Sanger sequencing was performed in these two exons, as described previously [18], using primers mentioned in Appendix A. The *MYOC* negative families thus identified were taken further for whole-exome sequencing.

### 2.3. Whole-Exome Sequencing and Analysis

Whole-exome sequencing was performed using Illumina’s Nextera Rapid Capture Expanded Exome kit for 46 samples. For the remaining eight samples, the TrueSeq exome enrichment kit was used due to the experiments carried out in two phases during the study (Appendix A). The library preparation was performed according to the manufacturer recommended protocol and 100 bp paired end reads were generated on HiSeq2000 (Illumina Inc., San Diego, CA, USA). The data generated for this study have been submitted in the NCBI Sequence Read Archive (SRA BioProject ID: PRJNA394051 and SRP113309).

The quality of sequencing reads was assessed using FastQC (https://www.bioinformatics.babraham.ac.uk/projects/fastqc/ (accessed on 1 May 2018)) and Trimmomatic [19]. The good-quality reads, with bases having a Phred quality score more than Q30 (lower quality reads from read ends were trimmed) were included in the analysis. The reads were aligned to the hg19 reference genome using BWA (BWA backtrack 0.7.4) [20]. The conversion of SAM to BAM and duplicate removal was performed using Picard (Picard Tools—By Broad Institute, 2018: http://picard.sourceforge.net, accessed on 13 March 2018) [21]. The local realignment around indels and base quality score recalibration (BQSR) was performed before calling variations using GATK’s Haplotypecaller [22]. High-quality variations were filtered according to GATK-defined parameters. These variations were annotated using ANNOVAR [23] and GEMINI [24]. One sample (II.4) from Family F_5 was excluded from the study due to low coverage (<25×).

### 2.4. Identification of Candidate Variations and Prioritization

Familial glaucoma, especially POAG, is generally known to follow an autosomal dominant mode of inheritance; but there are families which do not follow a recognizable pattern of inheritance and show considerable incomplete penetrance and variable expressivity. In the nine families included in our study, we identified the variations that segregated with the affected individuals (Appendix A), i.e., variants present in heterozygous or homozygous alternate form in cases but absent in unaffected family members. In families F_4 and F_9, we also used an autosomal recessive model of inheritance for data analysis, as it consisted of early-onset glaucoma cases.

The segregating variations hence identified were filtered based on allele frequency and functional prediction. The allele frequencies were identified using 1000G (1000 Genomes population) [25] and ExAc (Exome Aggregation Consortium) [26] datasets. The variations with alternate allele frequency >0.01% or that had an rsID in SNP database db138 were considered non-pathogenic.

The pathogenicity of the variations was analyzed based on the ljb26 database in ANNOVAR (prediction on the basis of evolutionary conservation and protein structure). The functional prediction for deleteriousness of a variation was on the basis of various scoring methods, including SIFT [27], PolyPhen2 [28], HDIV/HVAR, LRT [29], MutationTaster [30], MutationAssessor [31] and FATHMM [32] scores. A set of candidate single nucleotide variations (SNVs) were identified that were predicted to be deleterious by a minimum of three algorithms. This set of candidate variations were further prioritized using conservation score predictions with the help of CADD [33], GERP [34] PhyloP [35] scores and their link to glaucoma related genes/pathways. Apart from SNVs, the frameshift and non-frameshift Indels were analyzed and rare indels were selected in the study.

### 2.5. Known Gene Analysis

The whole-exome data from the familial samples were analyzed for non-synonymous variations in the known POAG genes, namely *OPTN, CYP1B1, WDR36, ASB10, NTF4, TBK1, IL20RB, LTBP2 and OPA1*. For PACG families, we analyzed for rare segregating deleterious variations in known associated genes, namely *MFRP, MTHFR, MMP-9, HSP70, PLEKHA7, COL11A1, PCMTD1-ST18, ABCC5, LTBP2, eNOS, EPDR1, CHAT, GLIS3, FERMT2 and DPM2-FAM102A.*

### 2.6. Genotyping of Prioritized Variations in a Validation Cohort

We selected 33 deleterious variations (16 POAG families + 17 PACG families) in candidate genes based on the strategy described above. These were genotyped in 960 cases and 576 controls. All of the 1536 DNA samples were diluted to 15 ng/uL and passed the standard quality checks. The samples were genotyped using iPLEX^®^ Gold genotyping reagents using standard protocol on Sequenom MS (TOF)—MassARRAY system. Primers used were designed using the Assay Designer software (Sequenom) (tabulated in Appendix A). This experiment was conducted according to manufacturer’s guidelines. The genotypes were automatically called by the Sequenom software Typer.

### 2.7. Immunohistochemistry (IHC)

IHC was performed in human retina for SIGLEC11 using anti-SIGLEC11 antibody ab106390 (Abcam, Cambridge, UK). For SIGLEC11 IHC, 4-micron thick sections were prepared from formalin-fixed, paraffin-embedded tissue block. All Immunohistochemistry steps were performed on the BondMax and utilized the Bond Polymer Refine Red Detection Kit (alkaline phosphatase chromogen; Leica Microsystems). Tissue sections were deparaffinized and Heat Induced Epitope Retrieval (HIER) performed on the instrument using Epitope Retrieval Solution 2 (EDTA-buffer pH8.8) at 98 °C for 20 min. All slides were incubated with the primary antibody SIG1 (1:250) for 15 min, post-primary AP for 15 min, polymer AP for 15 min, mixed red refine (alkaline phosphatase, red chromogen) for 10 min, and hematoxylin as counterstain for 5 min. The slides were then cover-slipped on Leica CV5030. Between incubations, sections were washed with Tris-buffered saline (bond wash solution).

### 2.8. Gene Interaction and Pathway Analysis

The candidate genes, in which we found rare deleterious variations segregating with the disease, were analyzed for enrichment in the biological pathways using WEB-based Enrichr [36,37]. Enrichment visualization analysis was performed using ClueGO and CluePedia [38] applications of Cytoscape [39]. In ClueGo settings, the GO biological processes, GO cellular components, KEGG, REACTOME and WikiPathways were selected to identify the pathways enriched by the candidate genes in the families. The *p*-value was set to 0.05. Interaction maps of candidate genes with known genes were plotted.

### 2.9. Expression Analysis

To further investigate the possible biological relevance of the candidate genes in glaucoma, publicly available expression datasets were analyzed.

For the single-cell analysis, the Single Cell Portal (https://singlecell.broadinstitute.org/single_cell, (accessed on 1 May 2022) was searched to identify studies involving tissues associated with glaucoma. We identified three relevant studies [40,41,42] (Appendix A), involving expression analysis of aqueous humor output pathways, human retina and ocular compartments. The processed data were downloaded and used to identify expression of candidate genes in cells of ocular tissues. The cell types with a higher percentage of cells expressing candidate genes in each dataset were analyzed by comparing the status of candidate genes in each cell type against all other cells (base mean) within the same study, and a *t*-test was used to calculate significance. The R package ggpubr was used to process the data [43].

For the differential expression analysis of the candidate genes at the level of whole tissue, the Gene Expression Omnibus (GEO) data repository was probed to identify glaucoma-related publicly available gene expression profiles. Search terms included “Glaucoma”, “POAG”, “Retinal Ganglion Cells” and “Trabecular Meshwork”. We identified 14 GEO series (GSEs, Appendix A) that had freely available data relevant to our study [44,45,46,47,48,49,50,51,52,53]. The raw files were retrieved using GEOquery [54] and analyzed using affy [55] and limma [56] packages of BioConductor in R. All the raw expression files (.cel) were processed using RMA normalization. A total of 205 genes with expression values in GEO datasets were analyzed for differential expression and checked for significance (*p*-value < 0.05) using a student’s *t*-test. Plots were created for genes significantly differentially expressed in the POAG and PACG families.

The entire study design and workflow is depicted in Figure 1 below.

## 3. Results

### 3.1. Candidate Gene Screening Reveals Most Families Do Not Carry Causal Variants in Known Genes

A total of 37 samples from 11 families were collected for the study. The individuals underwent ophthalmological examinations to determine glaucoma affection status. The mean age of the samples was 49 years, with the age range being 12 years to 80 years (Appendix A). *MYOC* exons 1 and 3 screening led to identification of known mutations in two families. A p.Gln48His [18] variant in family F_10 and p.Thr353Ile [57] variant in family F_11 were found to co-segregate with the phenotype. In F_8 (individual III.3), a novel p.Glu409Asp variant was found which did not segregate with the phenotype (Appendix A). Thus, 31 samples from nine families were taken forward for exome sequencing (Figure 2) after excluding F_10 and F_11. This comprised 16 individuals from five POAG families and 15 from four PACG families. Twenty sporadic cases of glaucoma were also included for whole-exome sequencing.

All familial samples were analyzed for possible deleterious variants in known Mendelian glaucoma genes (Appendix A). A total of 18 unique non-synonymous variations were identified in *OPTN, CYP1B1, WDR36, ASB10, LTBP2, OPA1, PLEKHA7, COL11A1, PCMTD1 and CHAT*. The variations in these genes were common benign polymorphisms with variations having allele frequencies of more than 1% in the ExAc database. A non-synonymous change in *TBK1* (NM_013254, exon17, c.G1839T, p.L613F) was identified in patients of family F_3 and had an allele frequency of 0.001 in ExAc (and 0.0075 in South Asians). This was predicted to be deleterious by SIFT but neutral by other prediction tools. A potentially deleterious variation (NM_001035254: exon7; c.G634C; p.D212H) was present in *FAM102A* in F_9. Among the ExAc populations, this Asp to His change was present exclusively in the South Asian population with an allele frequency of 0.0008364. These findings did not lead to further exclusion of families as the known gene variations in these families were either common or did not segregate with the disease in the relevant families.

### 3.2. Novel Candidate Genes Identified by Whole-Exome Sequencing

In the five families with POAG, we sequenced 16 individuals for the whole exome and identified 146 rare, deleterious, non-synonymous SNVs segregating with the disease. These variations were found in 144 genes. There were 44, 31, 41, 19 and 11 such SNVs in F_1, F_2, F_3, F_4 and F_5, respectively (Figure 2). Variations in *NSMAF* and *ASPM* were identified in cases from more than one POAG family. Similarly, for the four PACG families, we identified 166 non-synonymous, deleterious SNVs from 153 genes. There were 90, 12, 33 and 31 such SNVs in F_6, F_7, F_8 and F_9, respectively (Appendix A). Variations in *TNRC18*, *INSRR*, *DACH2*, *TUBA3E*, *KRT85*, *TUBA3C*, *HIST1H4F*, *KRT2* and *CYP26A1* were identified in at least two PACG families.

The exome data was also analyzed to identify the rare indels that segregated with the disease in the families. In POAG families we identified 47 frameshift indels or indels at splice sites, and 19 in-frame codon indels. These indels were located in 54 genes. The PACG families had a total of 4 in-frame and 15 rare frameshift indels, coming from 15 genes. (Appendix A).

From the data described above, we have identified a total of 352 candidate genes in familial forms of glaucoma. Interestingly, *KRT2* showed both deleterious SNVs and indel segregating with the disease. The SNV in *KRT2* was found in F_5 and the indel in the same gene was found in F_1.

These 352 candidate genes used for expression and network analysis are described below.

### 3.3. Single-Cell Transcriptomic Analysis Suggests Enrichment of Candidate Genes in Glaucoma-Related Tissues

We probed three single-cell studies to identify the status of expression of candidate genes in anterior segment tissues and retinal tissues. We identified multiple genes of interest with higher relative expression in glaucoma-related cell types; examples include *NEFM*, *SYT2* expression in *RGCs*, *AQP5* and *KRT3* expression in corneal epithelial cells and *CDH6* in ciliary muscle. The distribution of candidate genes in ocular tissue in all families is visualized as dotplots in Appendix A.

We analyzed the families on the basis of mutations in percentage of genes expressed in tissues of anterior segment, posterior segment or both. In the POAG, Families F_2 and F_3 showed enrichment in corneal epithelial cells and conjunctival cells, indicating a mechanism associated with anterior segment. On the other hand, two (F_1 and F_4) out of five POAG families had enrichment of the percentage of genes expressed in retinal ganglion cells. In PACG, three families had enrichment in the RGCs (F_6, F_7 and F_9) and Schwalbe line (F_6, F_7 and F_8). Other cells of interest include corneal epithelial cells, ciliary body cells and smooth muscle cells (Figure 3).

Differential expression analysis (expression profiles using whole tissue) of the candidate genes in glaucoma-related studies further strengthens their role in glaucoma. Appendix A heatmaps represent the differential expression status of the genes having SNVs and the indels in various GEO datasets. Differential expression was identified in 224 candidate genes in at least one GEO study (Appendix A).

### 3.4. Key Pathways Involved in Families

The pathway analysis of the candidate genes from POAG families identified post-chaperonin tubulin folding, extracellular matrix (ECM) organization and transcription factor regulation in adipogenesis as the top pathways in Reactome and WikiPathways. The top transcription factors enriched were ESR2 (ChEa 2016, ChIP-Seq MCF-7 Human) and FOXM1 (ChEa 2016, MCF-7 hg19) (Appendix A). An interaction map of candidate and known POAG genes is represented in Figure 4a. The ECM component forms a major hub, and other notable hubs include cell adhesion and negative regulation of epithelial cell proliferation

The key pathways in KEGG, WikiPathways and Reactome databases for the candidate genes in PACG families include ECM interaction and organization, and lipid metabolism (Appendix A). The interaction map of candidate and known glaucoma genes is represented in Figure 4b. Analysis of the significantly enriched pathways using cytoscape’s plugins ClueGO and CluePedia revealed the genes forming two major hubs related to the above pathways.

### 3.5. Prioritized Genes Harbor Variations in Sporadic Glaucoma Patients

After expression and pathway analysis of our wide set of candidate genes, we sought to follow a strict filtering criterion described in the method section of prioritizing the variants to further narrow our gene list. We selected a set of 33 SNVs to be validated further in an East Indian Glaucoma Cohort. After analyzing the genotyped data in 960 sporadic glaucoma cases and 576 controls, we found seven SNVs from seven genes exclusively in cases (Appendix A). Out of these seven genes, *CDH6, SRFBP1, FOXM1* and *AQP5* were identified from the POAG families, while *RGL3, ACACB and LAMA2* were found in the PACG families. Thus, our two-tier strategy of unbiased, whole-exome sequencing in familial samples and validating a prioritized set in a case-control cohort has led to identification of seven potential candidate genes for glaucoma.

To further strengthen this finding, we have also sequenced a subset of 20 sporadic singlet glaucoma samples and analyzed variants in these seven genes. We identified rare, possibly pathogenic variations in *AQP5, FOXM1, SRFBP1* and *ACACB* in the sporadic glaucoma cases that were not present in the families. The variations included p.P245H change (NM_001651: exon4, c.C734A) in *AQP5*, p.S741Y (NM_001243088: exon8, c.C2222A) in *FOXM1*, p.R54L (NM_152546: exon3, c.G161T) in *SRFBP1* and three variations in *ACACB* involving p.D430Y (NM_001093: exon7, c.G1288T), p.L961M (NM_001093: exon18, c.C2881A) and p.L1762I (NM_001093: exon37, c.C5284A)—these are summarized in Appendix A.

Among the validated genes with SNVs, the whole tissue differential expression analysis of *CDH6* (five out of six studies with at least -0.13 fold change), *SRFBP1* (two of three studies; at least -0.8 fold change), *RGL3* (two of three studies; at least -0.2 fold change) and *ACACB* (four of four studies; -0.2 fold change) shows a trend of downregulation, whereas *AQP5* (two of three studies; at least 0.3 fold change) and *LAMA2* (four of four studies; at least 0.14 fold change) show upregulation in studies (Appendix A heatmaps).

### 3.6. Specific Families with Interesting Findings

#### 3.6.1. POAG Family F_4: A family with Variable Age of POAG Onset

Family F_4 had four members sequenced for the whole exome. The proband (III.1) is a female with juvenile open angle glaucoma, which was detected at the age of 35 years. She also suffered from early-onset diabetes and had thyroid abnormalities. The mother (II.1) and aunt (II.4) of the proband, ages 66 and 60 years, had late-onset glaucoma (POAG). The mother of the proband also suffered from diabetes, thyroid and had high levels of serum cholesterol. Apart from the affected members of the family, a sample from an unaffected uncle of the proband was also obtained. The uncle (II.9), aged 52 years, did not suffer with any ocular or systemic disease. The exome sequencing and subsequent analysis led to identification of segregating variations in five genes that may be associated with the pathologies in the family (Figure 5A). The genes segregating by autosomal dominant mode of inheritance and subsequent status in the cohort study led to identification of SNVs in *SRFBP1* and *CDH6* and duplications in *SIGLEC11* (NM_001135163: Chr19: 50455586, c.1407_1428dup21) and *PAX4* (Chr7: 127251016, NM_006193, c.906_912dup7, p.Pro304fs) as the probable genes of interest. Below we have discussed each variation in detail.

We identified Serum Response Factor Binding Protein 1 (*SRFBP1*) as a candidate gene in the *GLC1M* locus [58]. A p.Glu12Gln change was present in *SRFBP1*. This variation was predicted to be deleterious by five algorithms. It also had high CADD (15.07) prediction and GERP++ (5.12) scores, implying conservation across species. It was present in 3 out of 120,640 alleles in the ExAC database. Two out of the three alleles were identified in the South Asian population, and one was from European (non-Finnish) population. In our case-control cohort, it was present in 2 of 937 cases and none of the 573 controls. Further, the exome analysis of 20 sporadic samples revealed an additional deleterious p.Arg54Leu change (NM_152546: exon3, c.G161) in sample S9. *SRFBP1* is located within juvenile open angle glaucoma loci *GLC1M* on chromosome 5q [58]. This gene is also present within the fine-mapped region of this locus [59] (Figure 5B). It is interesting to note that the family under investigation also has a JOAG member and has a South East Asian ancestry, as in the original family where this locus was mapped. Other than the glaucoma loci, this gene was also present in butterfly-shaped macular dystrophy locus 5q21.3-q33.2 [60]. It is also interesting to note that both the identified variations in *SRFBP1* (p.Glu12Gln in family and p.Arg54Leu in a singlet case) were present in the SRF binding domain of the gene, and both variations show high conservation across species (Figure 5C). A 100 kb region was identified around the variation for haplotype analysis. We selected the informative heterozygous SNPs in this region. A haplotype was built using the genotype information from the four members of family F_4. Four different haplotypes were identified in the family using SNP markers around the variation based on the IBD analysis. One of the haplotypes segregated with the glaucoma phenotype (Figure 5D).

The heterozygous in-frame 21 bp duplication in *SIGLEC11* near the transmembrane region was resolved by sequencing the TA clones obtained from patient DNA (Appendix A). A screening of 250 additional cases identified the same duplication in another patient (GL_1025). We also found a p.Leu559Pro change in *SIGLEC11* (Chr19: 50455599; rs145904036, ExAc frequency: 0.0002) in one other patient (GL_932). Both of these changes were absent in the 258 controls sequenced. Immunohistochemistry for SIGLEC11 in the human eye sections revealed its localization in the corneal epithelium but was absent in stroma and endothelium. In the ciliary body and pars plana, the staining was much stronger in pigmented epithelium, with some staining observed in non-pigmented epithelium. SIGLEC11 was also located in the dilator muscles of the iris. The staining was absent in the retina (Figure 5E).

*PAX4* is a known candidate gene for diabetes. Mutations in this gene have long been associated with type 2 diabetes [61,62] and the frame shift change may play an important role in the diabetic phenotype in this family.

As an alternative analysis strategy, we also analyzed the data to identify possible genetic variations related to the JOAG (more severe) and POAG phenotypes (less severe) in the family. Thus, we used a model where the JOAG patient (III.1) would reveal a homozygous alternate allele, while the POAG patients (II.1, II.4) are heterozygous for the alleles and the unaffected member (II.9) has the homozygous reference alleles. A proline-to-serine change in *NGB* (NM_021257: exon2, c.C175T, p.Pro59Ser) was the only deleterious change identified in the family using this model.

#### 3.6.2. PACG Family: Genetic Basis of Co-Occurrence of PACG and Hypercholesterolemia in F_9

Family F_9 had consanguineous marriages (mating relationship) within the family (Figure 6A). Three members of this family are affected with angle closure and hypercholesterolemia. The proband had features of xanthelasma patches in both palms and upper lids of both eyes (Appendix A). The mother and younger sibling of the proband were diagnosed with hypercholesterolemia and are on medical treatment, with no evident xanthelasma in the body. All three females in the family had associated angle closure disease, which was, however, severe in the proband, with closed angles in all quadrants and advanced disc and field damage at presentation. The mother (III.4) and younger sister (IV.2) had only PAC with no disc or field changes and two quadrants of angle closure. The twin brother (IV.3) of the younger sister and father were normal, with no hypercholesterolemia or angle closure disease.

In this family, we applied autosomal dominant and autosomal recessive (homozygous alternate variations are reported in early-onset glaucoma) models to identify the variations involved with PACG and hypercholesterolemia. Based on the pedigree structure and the above analysis, variations in *RGL3, ACACB, LDLR* and *PCSK9*, among others, co-segregated with the phenotypes and can implicate the pathologies in F_9.

The variations in *ACACB* followed the autosomal dominant model. The p.Pro474Leu (chr12:109614052; NM_001093: exon8: c.C1421T) occurs in homozygous reference form in unaffected (III.7, IV.3) and heterozygous form in the affected (III.4, IV.1, IV.2) members of the family. This mutation is reported in 3 of 122,945 alleles in the ExAc database. In our glaucoma cohort, it was present in 1 of 927 cases and in none of the 557 controls.

The mutations in RGL3 and LDLR are present in heterozygous form in the parents (III.7, III.4) and homozygous alternate in the three siblings (IV.1, IV.2, IV.3). RGL3 has a missense mutation leading to Thr436Ilu change (chr19:11513153, NM_001035223, NM_001161616; exon11, c.C1307T). This variation is reported to be present in 9/118,414 alleles in the EXAc database (8/14,614 alleles in the South Asian population, 1/11,532 in the Latino population). The variation screening in our cohort revealed its presence in three cases and no controls. A four bp (C/CATCA) frameshift insertion (p.Asp100fs NM_001195802, p.Asp221fs NM_001195798,) in LDLR was identified in the family following a similar inheritance pattern.

As only the proband suffered a severe form of both the diseases, we also looked for variations present in this case only and not in other members. We identified an in-frame insertion of leucine in the PCSK9 signal peptide. A known polymorphism, p.Leu23dup, was present in heterozygous (L9/L10) form in four family members other than the proband, in whom it is present in an alternate homozygous (L10/L10) form. This multiallelic variation incorporates an additional leucine to a tandem repeat of nine leucines in the signal peptide region. The literature suggests this mutation to be a modifier and it seems to be an important factor in the early onset and severity of the diseases in the context of this family.

We are of the opinion that in this family, LDLR and PCSK9 may not only cause hypercholesterolemia but may also play an important role in glaucoma. Genes identified in this family interact with the known glaucoma genes involved in cholesterol dysregulation (Figure 6B). ABCA1, APOE, CAV1, CAV2, GHRL, GHSR, ESR1 and ESR2 [63] are the glaucoma-linked genes that have a role in lipid metabolism.

Further from our single-cell transcriptome analysis (Figure 6C) we identified the expression of the above genes in important glaucoma-related tissues. LDLR is particularly interesting here as it has high expression in the RGCs and cells of aqueous humor outflow pathways, indicating its contribution to glaucoma along with the hypercholesteremia phenotype.

## 4. Discussion

Glaucomatous neurodegeneration, like most complex genetic diseases, has both Mendelian (single gene) and multifactorial components. The genotype-to-phenotype correlation largely depends on the penetrance of the contributing alleles. Typically, identification of high-penetrant rare alleles in familial forms of the disease will shed deeper insights into the disease mechanism than statistical association of low-penetrant common alleles. This study describes a strategy to identify rare variations that would contribute to glaucoma susceptibility in *MYOC*-negative families. The exome sequencing and analysis was carried out for four PACG and five POAG families. The analysis of known glaucoma genes led to the identification of non-synonymous and synonymous variations. All of these variations were commonly found in the population, thereby negating their chance of causing glaucoma.

The variations in novel genes segregating with the disease in the families were filtered. A total of 352 probable candidate genes that may be involved in the disease pathology were identified in the four PACG and five POAG families. Among the candidate genes, 33 most potential single nucleotide variations were screened in an independent validation cohort consisting of 960 cases and 576 controls.

Prioritized gene variants in our glaucoma case-control cohort supported possible causal roles in four genes for POAG (*AQP5, FOXM1, SRFBP1* and *CDH6*) and three genes in PACG (*LAMA2, ACACB* and *RGL3*). Based on our findings of mutations in multiple samples in these genes and literature support, it is likely that these genes contribute to glaucomatous neurodegeneration beyond the Indian subcontinent.

Analysis of 20 independent glaucoma cases revealed additional deleterious variations in four (out of seven) validated genes. The validated genes in glaucoma families included SRFBP1. SRFBP1 is present in a known glaucoma linkage locus: GLC1M. These genes provide novel connections for the involved biological pathways for future studies. Further, pathway analysis of all 352 candidate genes revealed ECM organization and lipid metabolism to be significantly enriched for the potentially pathogenic variations in our cohort.

The single-cell analysis also identified enrichment in epithelial cells. Negative regulation of epithelial cell proliferation forms a hub in the network analysis, indicating the importance of epithelial cells in glaucoma. Together, the candidate genes contribute to larger networks, like extracellular matrix remodeling, inflammation and lipid metabolism, which already has a reported role in glaucoma. The known glaucoma pathways include eye development, lipid metabolism, inflammation, ECM organization, oxidative stress and senescence, etc. [63,64,65,66]. The ECM organization is implicated not only in trabecular meshwork, but it is also overrepresented in the lamina cribrosa, therefore the effect of potential pathogenic variations in the genes can impact functions of both anterior and the posterior parts of the eye. In PACG, lipid metabolism seems to play an important role. Though a high lipid level is not a risk factor for glaucoma, there are studies that have identified a relation between the two. Ultra-high-resolution metabolomics identified significant alteration in lipid-metabolism-related processes in POAG [67]. Dyslipidemia was found to be significantly associated with a high intraocular pressure [68], but not glaucoma. In another study, rabbits fed with a high-cholesterol diet resulted in features similar to glaucoma [69]. The rabbits suffered from a primary form of glaucoma as there were developmental defects in the filtration angle. Further, they report that the lesions in the eye tissues caused by hypercholesterolemia may have a role in deterioration of the disease. In addition, short-term use of statins, cholesterol lowering drugs, was associated with a decreased incidence of glaucoma [70]. There are several genes related to lipid metabolism that have been associated with glaucoma. These genes include *ABCA1* [71,72], *APOE* [73], *CAV1/CAV2* [74], *OPA1* [75] and *CYP1B1* [76]. These findings point to the role of cholesterol dysregulation in glaucoma and must be further probed.

Glaucoma can be defined as loss of retinal ganglion cells. However, underlying this simple definition lies tremendous complexity. Glaucoma is a genetically heterogeneous disease, with multiple factors responsible for the manifestation of the disease. The large number of candidate genes identified from the linkage and association studies converge onto a few pathways. These include eye development, lipid metabolism, inflammation, ECM organization, oxidative stress and senescence [63,64,65,66]. Most of the glaucoma-associated variations ultimately make the RGCs more prone to apoptosis [77,78,79,80]. The genes identified in our study, as potential contributors to glaucoma, align with the known pathways and contribute further to our understanding. The neuroinflammation, via the glial cells, of the optic nerve head is considered to be critical in disrupting the homeostasis of the RGCs [81,82]. SIGLEC11, identified as a candidate gene in the study, may be a key player in maintaining such homeostasis, as it has a known neuroprotective function and is expressed in retinal microglial cells. The ECM organization in the anterior chamber tissues, especially in the trabecular meshwork, play an important role in the maintaining an optimum IOP [83,84]. The likely pathogenic variations in CDH6, SRFBP1 and LAMA2 may assist in disorganization of the ECM components and cytoskeleton, leading to hypertension. The pathway enrichment has shown to vary with the ocular tissues associated with glaucoma [63]. The ECM organization is implicated not only in trabecular meshwork, but it is also overrepresented in the lamina cribrosa; therefore, the effect of potential pathogenic variations in CDH6, SRFBP1 and LAMA2 can impact functions of both the anterior and posterior parts of the eye. Nuclear factor κB (NF-κB) was identified to be the most important upstream regulator of candidate genes expressed in the retina and optic nerve head astrocytes [63]. RGL3, in association with OPTN and TBK1, plays an important role in regulating the autophagy and NF-κB signaling in these tissues [85].

It is common for two or more complex diseases to co-occur. This can be due to high prevalence of conditions and to underlying genetic factors contributing to both conditions. Genetic interactions and multifunctional proteins play an important role in all types of complex diseases [86]. The study of inverse co-morbidities also provides important clues to disease biology [86,87]. Glaucoma and an elevated IOP are associated with a number of systemic diseases. These include diabetes mellitus, cancer, hyperthyroidism, hypertension and hyperlipidemia, to name a few [88] [70]. Though the role of diabetes and hypertension in glaucoma is debatable, several studies reported them as co-occurring conditions. In our study, we identified deleterious frameshift duplication in PAX4 in a family having both diabetes and POAG. PAX4 is one of the important candidate genes of diabetes [61] and may be involved in regulation in the optic nerve tissue [89]. Lipid metabolism has been identified as one of the most important pathways in glaucoma [63]. We identified potentially deleterious variations in *ACACB*, *LDLR* and *PCSK9* in a family with co-occurrence of hypercholesterolemia. ACACB help catalyse conversion of acetyl-CoA to malonyl-CoA, and it is an important regulator of fatty acid oxidation [90,91]. *LDLR* and *PCSK9* are the known genes for familial hypercholesterolemia [92,93]. PCSK9 prevents the recycling of *LDLR* and helps in its degradation [94]. Due to the practical limitations and ethical concerns of following up with individual patients, it was not possible to delineate genotype–phenotype correlations for individual genes in cases where more than one disease co-occurs.

Further studies should focus on functional investigations and unraveling the molecular mechanisms of these genes. Studies using multidimensional approaches will be helpful for a deeper understanding of the disease mechanism. The complex gene networks can serve as important focus points for targeted therapy. The BDNF- [95,96] and anti-TNF-α [97]-based treatments are already showing promising results. Molecules targeting other glaucoma-associated networks, like ECM remodeling, inflammation and RGC apoptosis, should be investigated. Drugs targeting specific glaucoma-associated pathways along with patients’ genetic information would be a great step in personalized and precision medicine in the field of glaucoma.

## Figures and Tables

**Figure 1 genes-14-00495-f001:**
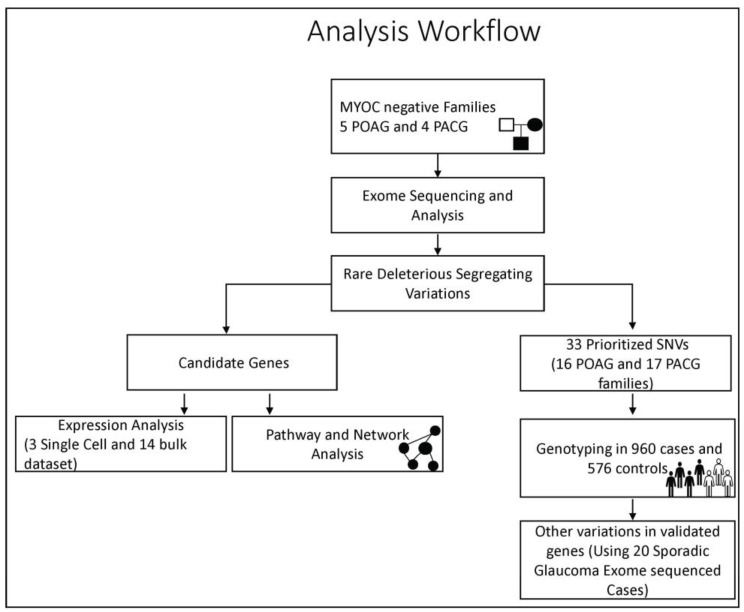
**The workflow depicting the study design and analysis strategy adopted in the study**.

**Figure 2 genes-14-00495-f002:**
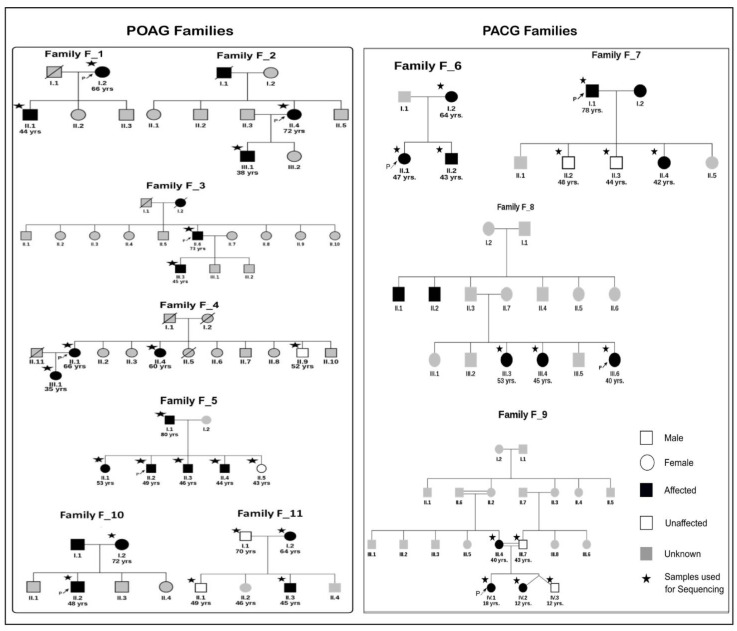
**The pedigrees of seven POAG families and four PACG families used for exome sequencing.** All individuals in white symbols have tested negative for glaucoma. The individuals in gray represent members with disease status unknown. Other symbols represent usual pedigree nomenclature. The arrow represents the proband and star * indicates the samples used for exome sequencing (Except families F_10 and F_11 were not exome sequenced as they had known *MYOC* mutations).

**Figure 3 genes-14-00495-f003:**
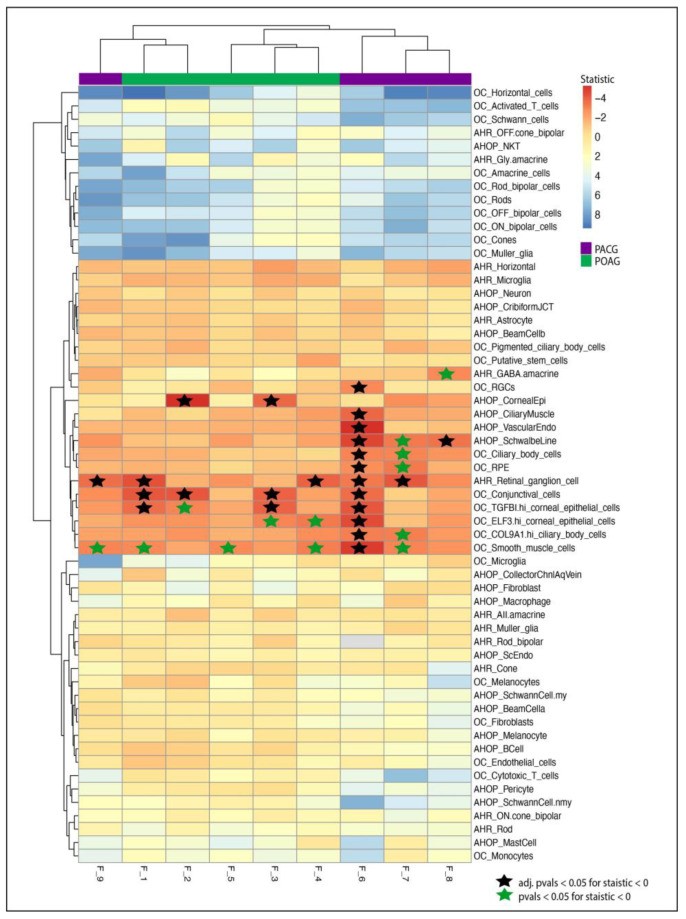
**Heatmap showing t statistic for candidate genes expressed in eye tissue.** Heatmap and clustering using the t statistic from the *t*-test of each cell type against all cell types from a study. Negative statistic (Red) represents enrichment of percent cells expressing candidate genes with respect to all tissues. The stars represent *p* value < 0.05 in tissues with negative statistic. The black stars represent adjusted *p* values (Bonferroni) less than 0.05. Blue stars represent significant *p*-values before adjustment.

**Figure 4 genes-14-00495-f004:**
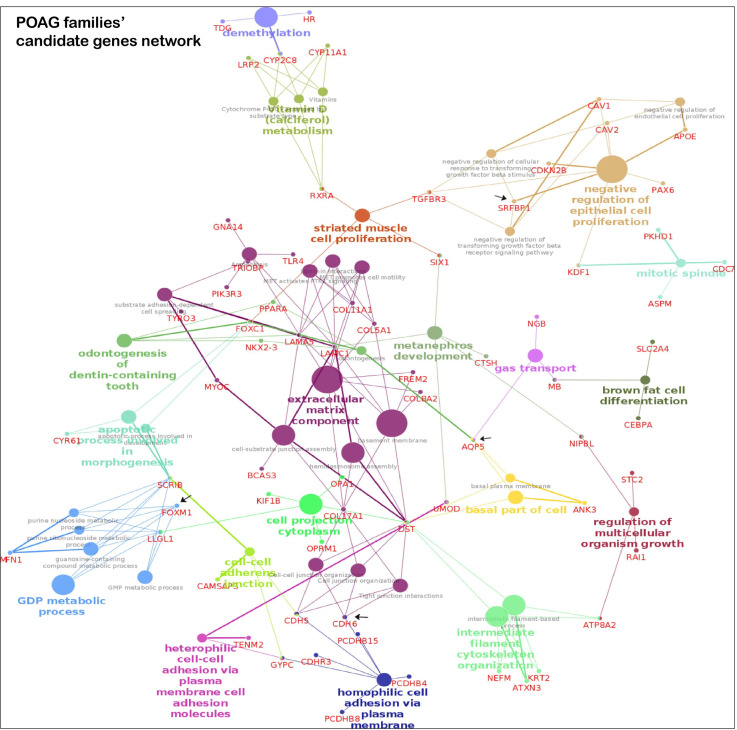
**Pathway enrichment from POAG/PACG candidate genes.** The interactions of known and candidate genes are represented. The node represents the GO, KEGG, WikiPathways and REACTOME items. The size of the node represents the significance; smaller nodes have higher and larger nodes have lower significance. The highlighted pathway is labeled according to the highest significance. The color of the nodes is visualized on the basis of different functional groups. Panel A is the genetic network map from candidates genes of POAG families and panel B is the same from the PACG families.

**Figure 5 genes-14-00495-f005:**
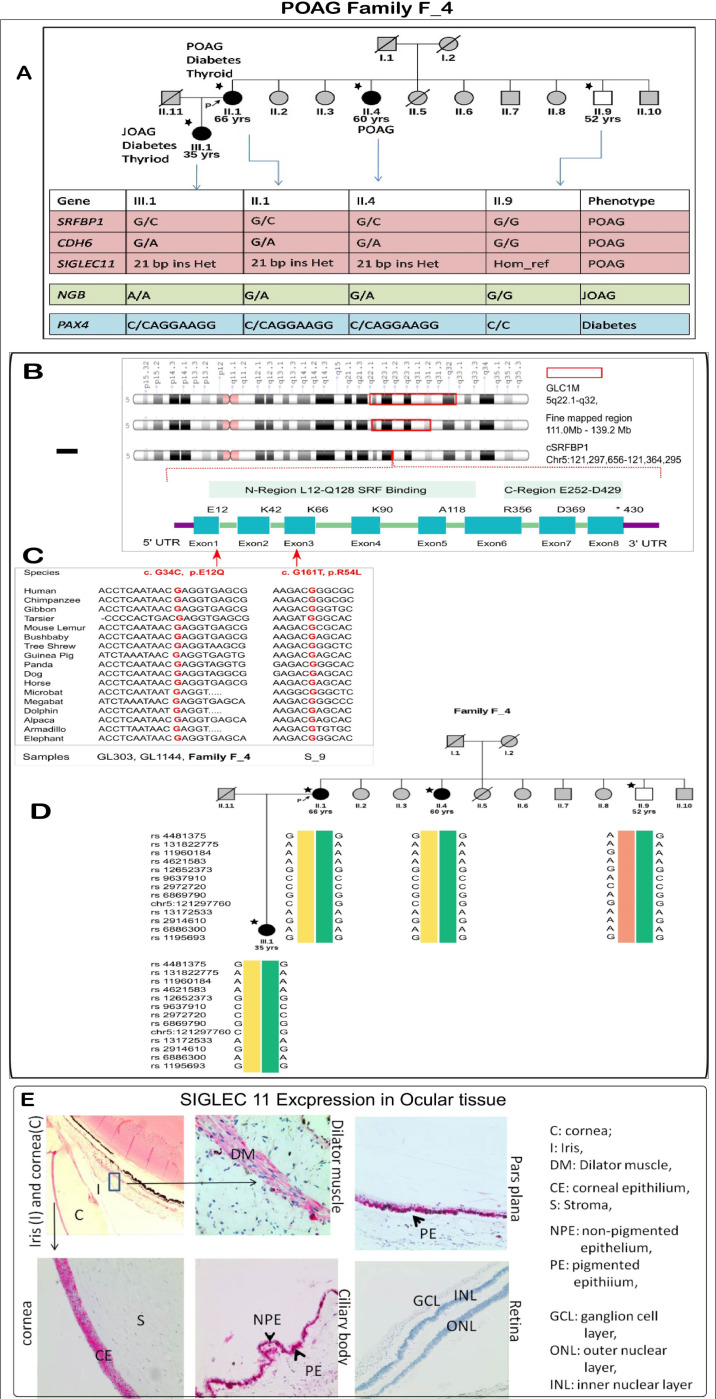
**Detailed analysis of a POAG family**. (**A**) Figure showing pedigree of the members of F_4 POAG family and the probable genotypes identified from our study that may be associated with the disease. The gene variations in pink and blue colors represent heterozygous variation in cases and homozygous reference in healthy. The gene variations in green represent homozygous alternate in JOAG, heterozygous variation in POAG and homozygous reference in healthy. Candidate gene SRFBP1: (**B**) A representation of the GLC1M loci (region first reported: 5q22.1-q32) of POAG. A fine-mapped region on the chromosome is also drawn. GLC1M loci harbor the candidate gene SRFBP1. The gene consists of eight exons and two functional regions, one of them being the SRF binding region. (**C**) The two variations identified in our cohort and their conservation across species. Both the changes are present in the SRF binding region (**D**) The haplotypes identified by the SNPs around the p.E12Q variation. The yellow haplotype segregates with the POAG phenotype in the family. The individuals marked with a star (*) underwent whole exome sequencing (**E**) Expression of Siglec11 in the human eye: Immunohistochemistry of SIGLEC11 revealed its expression in dilator muscles of the cornea, epithelium of the ciliary body and pars plana.

**Figure 6 genes-14-00495-f006:**
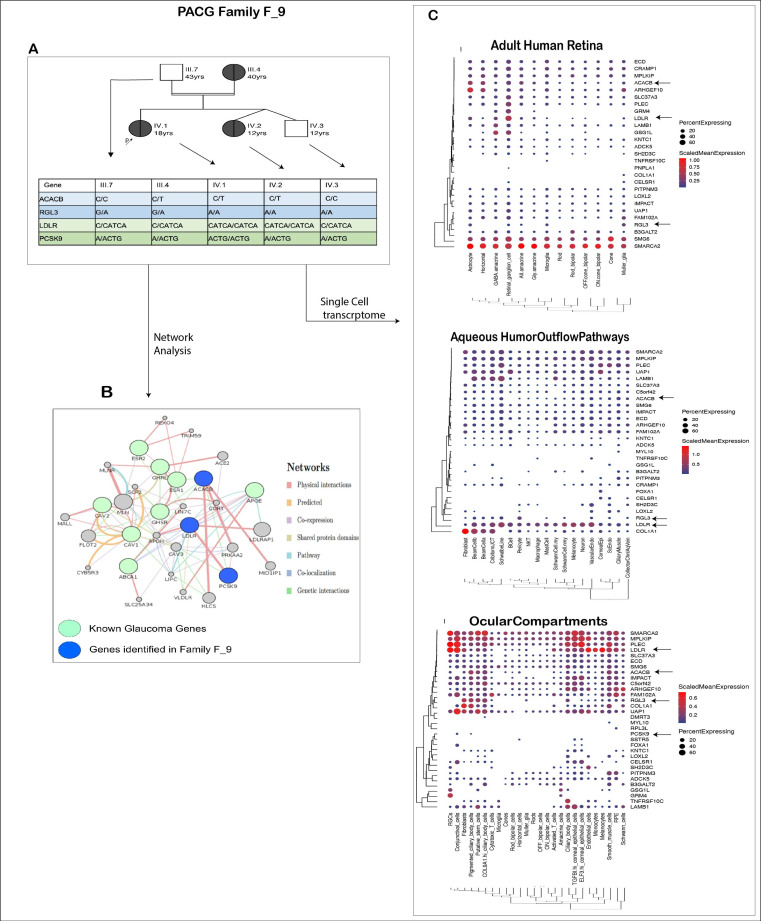
**Detailed analysis of PACG family.** (**A**) Figure showing pedigree of the members of F_9 PACG family and the probable genotypes identified from our study that may be associated with the disease. (**B**) Interaction of the known glaucoma genes that are associated with lipid metabolism (green circles) with the candidate genes ACACB, LDLR and PCSK9 identified in family F_9 (blue circles). Indirect interactions are also observed via other genes (gray) involved in the lipid metabolism and cholesterol pathways. The edges represent the type of interaction in the network, as is represented in the right panel. (**C**) Single-cell expression analysis of candidate genes in F9 family. The black arrows represent the above-discussed candidate genes in the family.

## Data Availability

The data generated for this study has been submitted in the NCBI Sequence Read Archive (SRA) under BioProject ID: PRJNA394051 and SRP113309.

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
