# Peer review of "Whole Exome Sequencing Reveals Novel Candidate Genes in Familial Forms of Glaucomatous Neurodegeneration"

_genes, 2023, doi:10.3390/genes14020495_

Round 1
Reviewer 1 Report
Reviewer comments to authors
In this work, Narta and colleagues present a sequencing-based work to identify novel pathways in two subtypes of Glaucoma: primary open angle glaucoma (POAG), and primary angle closure glaucoma (PAGG). The conducted research is clearly explained, with methods clear and descriptive and easy follow-up on the results. The build-up of the research went from description of well-know associated genes to conduct whole exome sequencing to correlate these two diseases with novel genes, and therefore novel pathways. The authors also took into account other pathologies present in the individuals of study, allowing them to discard some variants in genes associated with other pathologies (e.g. PAX4 is associated with diabetes). However, despite the high-quality work, there are still some aspects that should be improved before final
approval. These aspects are divided into mayor and minor comments:
Mayor comments:
- The authors focused their research on coding region or exosomes, however they did not explore intronic areas that can also harbour mutations that cause splicing aberrations or event the degradation of the mRNA leading to disruption of the protein translation. Which relevance has these kind of intronic mutations in glaucoma? I would suggest to include a statement or an explanation about why intronic areas were not considered.
- POAG families: As it was indicated before the authors nicely related some of the novel genes with other pathologies present in the individuals of analysis. However, only PAX4 is clearly related to the diabetes. However, other three genes (CDH6, SIGLEC11 and SRFBP1) that were also identified in a diabetic/thyroid/hypercholesterolemia affection context. The authors focused in the pathogenicity of the SNV of these genes, but they do not indicate if these genes are associated with other diseases, especially the ones present in the individuals of study. The authors should also add this aspect. Otherwise, the discussion is a bit disbalanced in this aspect.
Minor comments.
- Some of the main figures need to be improved to be readable.
Figure 3. Change the colour of one of the starts, black and dark blue are too close
to be properly differentiated. Green could be an alternative to blue.
Figure 4. Divide it in two main figures, in order to do them bigger.
Panel A. All the pathways that are written with smaller font size and in grey
colour are not readable (even with a big screen and big zoom). The author
must adapt the figure in order of all the displayed information can be
readable, otherwise provided information must be adapted.
Panel B. All the pathways written in grey are not readable, but also the
spot of the purple dots is to readable at all. Modify the figure accordingly.
Figure 5. Too small, make it bigger or divide the information in different figures.
Figure 6 Panel C. This panel is not readable. I would suggest to re-adapt the figure
to portrait distribution and elongate this panel.
There are some typos and italics missed along all the text. Here I include a list of the things I noticed, but I strongly encourage the authors to do a deep-revision.
 Line 53-54: (...) ‘ the patients remained unaccounted for by mutations (...). One of the prepositions may be removed.
 Line 236: ‘A p.Gln48His[18] variant (...)’: A space should be added between His and the reference.
 Line 262: Remove the extra dot of the end of the sentence.
 Line 267: Refer again to figure 2 after ‘ (...) such SNVs in F_1, F_2, F_3, F_4 and F_5, respectively (...)’. The figure was indicated long time before, indicating here again can help reader to connect information.
 Lines 279 and 280: KRT2 should be in italics due to it is referring to the gene.
 Lines 287-289: All genes (NEFM, SYT2, AQP5, KRT2 and CDH6) must be written in italics due to they are referring to gene expression.
 Lines 297-296: For consistency all families must be written in the same way (F_X) so correct F2 for F_2; F3 for F_3; F1 for F_1; F4 for F_4 and F7 for F_7.
 Line 341: Double space between ‘PAGG families.’ and ‘Thus, our (...)’, remove the double space.
 Lines 348-349: Again genes must be written in italics (AQP5, FOXM1, and SRFBP1).
 Lines 372-373: See previous comments, genes must be written in italics (
SRFBP1, CDH6, SIGLEC11, and PAX4)
 Line 376: GLC1M and SRFBP1 must be written in italics
 Line 431: Double space between ‘in human eye.’ and ‘Immunohistochemistry’. Remove the double space.
 Line 449: ‘and’ must be written without italics.
 Line 456-457: RGL3 and LDLR genes must be written in italics.
 Line 476: All genes of this line must be written in italics.
 Line 517: SRFBP1 and GLC1M must be written in italics
 Lines 541-542: All the gene listed in both lines must be written in italics
Reviewer 2 Report
In this manuscript Narta and al., describe the identification, by WES of a rare mutations with high penetrance in novel genes and gene-networks in familial forms of primary open angle glaucoma (POAG) and primary angle closure glaucoma (PACG).
Rare, deleterious SNVs in AQP5, SRFBP1, CDH6 and FOXM1 from POAG families and in ACACB, RGL3 and LAMA2 from PACG families were found exclusively in glaucoma cases. AQP5, SRFBP1 and CDH6 also revealed significant altered expression in glaucoma in expression datasets.
This is an interesting work, which show that pathway analysis of candidate genes revealed enrichment of extracellular matrix organization in both POAG and PACG. While I agree with the approach and study is of interest, I have few suggestions for authors to improve the paper.
- Some of identified genes for POAG (AQP5, FOXM1, SRFBP1 and CDH6) and PACG (LAMA2, ACACB and RGL3) are implicated in some pathologies, you have verified if the patient presents any clinical symptoms related to these genes?
- Line 47: add space between “years” and “[5]”
- Line 52: add point after “loci [8]”
- Line 114: add space between “method” and “[13]”
- Line 152: Why you don’t refer to GnomAD exome when you filtered on allele frequency?
- Line 160 and 164: add space before the reference number: “SIFT[27], PolyPhen2[28], HDIV/ HVAR, LRT[29], MutationTaster [30], MutationAssessor[31] and FATHMM[32]” and “CADD[33], GERP[34], PhyloP[35]”
- Line 279 and 280: KRT2 in italic, and also in all text same genes are not in italic (ex: line 348)
- Line 459: delete “chr19:11513153” it’s a repetition
